# A Novel Role for Plasminogen Activator Inhibitor Type-2 as a Hypochlorite-Resistant Serine Protease Inhibitor and Holdase Chaperone

**DOI:** 10.3390/cells11071152

**Published:** 2022-03-29

**Authors:** Jordan H. Cater, Noralyn B. Mañucat-Tan, Demi K. Georgiou, Guomao Zhao, Irina A. Buhimschi, Amy R. Wyatt, Marie Ranson

**Affiliations:** 1Illawarra Health and Medical Research Institute, University of Wollongong, Wollongong 2522, Australia; jcater@uow.edu.au; 2School of Chemistry and Biomolecular Science, University of Wollongong, Wollongong 2522, Australia; 3Flinders Health and Medical Research Institute and College of Medicine and Public Health, Flinders University, Bedford Park 5042, Australia; noralyn.manucattan@flinders.edu.au (N.B.M.-T.); demi.georgiou@flinders.edu.au (D.K.G.); 4Department of Obstetrics and Gynaecology, University of Illinois at Chicago College of Medicine, Chicago, IL 60611, USA; lisazhao@uic.edu (G.Z.); irina@uic.edu (I.A.B.)

**Keywords:** plasminogen activator inhibitor type-2, SERPINB2, hypochlorite, protein misfolding, preeclampsia, amyloid beta peptide

## Abstract

Plasminogen activator inhibitor type-2 (PAI-2), a member of the serpin family, is dramatically upregulated during pregnancy and in response to inflammation. Although PAI-2 exists in glycosylated and non-glycosylated forms in vivo, the majority of in vitro studies of PAI-2 have exclusively involved the intracellular non-glycosylated form. This study shows that exposure to inflammation-associated hypochlorite induces the oligomerisation of PAI-2 via a mechanism involving dityrosine formation. Compared to plasminogen activator inhibitor type-1 (PAI-1), both forms of PAI-2 are more resistant to hypochlorite-induced inactivation of its protease inhibitory activity. Holdase-type extracellular chaperone activity plays a putative non-canonical role for PAI-2. Our data demonstrate that glycosylated PAI-2 more efficiently inhibits the aggregation of Alzheimer’s disease and preeclampsia-associated amyloid beta peptide (Aβ), compared to non-glycosylated PAI-2 in vitro. However, hypochlorite-induced modification of non-glycosylated PAI-2 dramatically enhances its holdase activity by promoting the formation of very high-molecular-mass chaperone-active PAI-2 oligomers. Both PAI-2 forms protect against Aβ-induced cytotoxicity in the SH-SY5Y neuroblastoma cell line in vitro. In the villous placenta, PAI-2 is localised primarily to syncytiotrophoblast with wide interpersonal variation in women with preeclampsia and in gestational-age-matched controls. Although intracellular PAI-2 and Aβ staining localised to different placental cell types, some PAI-2 co-localised with Aβ in the extracellular plaque-like aggregated deposits abundant in preeclamptic placenta. Thus, PAI-2 potentially contributes to controlling aberrant fibrinolysis and the accumulation of misfolded proteins in states characterised by oxidative and proteostasis stress, such as in Alzheimer’s disease and preeclampsia.

## 1. Introduction

PAI-2, encoded by the *SERPINB2* gene, is best known as an inhibitor of fibrinolysis [1], efficiently inhibiting both urokinase plasminogen activator (uPA) and tissue plasminogen activator (tPA) [2] and mediating the rapid disposal of cell-surface-bound uPA and tPA via LRP1-mediated endocytosis in vitro [3,4]. PAI-2 levels in biological fluids increase dramatically in pregnancy and in response to inflammatory stimuli [1,5,6]. However, the biological importance of PAI-2 activity is still unclear, because under normal conditions, PAI-1 is a more abundant and efficient inhibitor of uPA and tPA in comparison to PAI-2 [2,7]. It has been proposed that in pregnancy PAI-2 protects against premature placental separation and maintains homeostasis by controlling proteolysis at the placenta [8]. Alternatively, it has been suggested that PAI-2 is a multifunctional protein that modulates T-cell responses [6] and can act as a holdase-type chaperone that stabilises misfolded proteins, limiting their aggregation within cells and in the biological fluids surrounding them [9]. In addition to being expressed by placental trophoblasts, PAI-2 is a stress protein that is strongly inducible in activated monocytes/macrophages and differentiating keratinocytes, fibroblasts and endothelial cells [1,5,6]. It has also been suggested that PAI-2 may have uniquely evolved to inhibit fibrinolysis during inflammation when elevated levels of biological oxidants result in the inactivation of PAI-1 [10,11]. In vivo, PAI-2 is present predominantly as a non-glycosylated intracellular form (PAI-2^NG^; 47 kDa), with only low levels of glycosylated PAI-2 (PAI-2^G^; 60 kDa) secreted into biological fluids due to an inefficient secretion signal [12,13]. However, PAI-2 levels in biological fluids increase dramatically in pregnancy and in response to inflammatory stimuli [1,5,6].

Owing mainly to the ease of recombinant protein production in prokaryotic systems, the vast majority of previous studies have focused on PAI-2^NG^, and as a result, very little is known about the corresponding properties of PAI-2^G^. This is important to address, because glycosylation can contribute to the chaperone activity of proteins [14,15] and protect proteins from damage induced by oxidants [16,17]. The purpose of this study was therefore to examine the effect of (i) glycosylation and (ii) exposure to hypochlorite (an oxidant and chlorinating agent generated during inflammation and at the placenta during pregnancy [18]) on the protease inhibitory and holdase-type chaperone activities of PAI-2. Furthermore, reduced PAI-2 expression is associated with the pregnancy-specific disorder preeclampsia [19,20,21], which is pathologically characterised by inflammation, fibrinolytic disturbances and protein misfolding inclusive of the amyloid beta peptide (Aβ) [22]. We therefore examined PAI-2 localisation relative to amyloid precursor protein (APP; from which Aβ is derived) in the human placenta as proof-of-principle that PAI-2 potentially contributes to extracellular protein homeostasis in the placenta in vivo.

## 2. Materials and Methods

### 2.1. Materials

Human recombinant PAI-1 was from Molecular Innovations (HPAI-R76E-I91L, Novi, MI, USA). Human recombinant wild-type PAI-2^G^ (~60 kDa; produced in CHO cells) and PAI-2^NG^ (~47 kDa; produced in *E. coli*) were from Biotech Australia (Roseville, Sydney, Australia). Human synthetic amyloid beta 1–42 peptide (Aβ_1–42_) was from Anaspec (Fremont, CA, USA). Unless stated otherwise, all other reagents were from Sigma-Aldrich (Sydney, Australia).

### 2.2. Protein Determination

Protein determinations were carried out by measuring the absorbance at 280 nm (A280) of solutions using a Nanodrop 2000C (Thermo Fisher Scientific, Melbourne, Australia). PAI-2 concentration was estimated assuming 0.1% Absorbance (1 mg/mL) = 0.92, and assuming a molecular weight of 47,000 and 60,000 g/mol for PAI-2^NG^ and PAI-2^G^, respectively.

### 2.3. Hypochlorite (NaOCl) Treatments

PAI-2^G^, PAI-2^NG^ or PAI-1 (0.25 mg/mL in phosphate-buffered saline (PBS)) were incubated for 2 h at 24 °C with 0–200 µM NaOCl. NaOCl-treated proteins were extensively dialysed to remove unreacted OCl- prior to being used in assays. Treatment with 200 µM NaOCl resulted in a 62 ± 27% and 75 ± 18% reduction in reactive thiols (cysteines) in PAI-2^G^ and PAI-2^NG^, respectively, as assessed using Ellman’s test colorimetric assay adapted from GoldBio Technology (TD-P Revision 3.0) and Uptima (UP01566H). Briefly, proteins were prepared at 200–330 µM in dithio-bis 2-nitrobenzoic acid (DNTB) solution (100 µM DNTB, 100 mM Tris buffer, pH 8.0), and the absorbance of the samples was measured at 412 nm. The extinction coefficient of DNTB (13,600 M^−1^ cm^−1^) was used to calculate the thiol concentration of the protein samples from a standard curve generated using N-acetyl-L-cysteine (Sigma-Aldrich).

### 2.4. Size Exclusion Chromatography (SEC)

PAI-2^G^ or PAI-2^NG^ were analysed by SEC at a flow-rate of 0.3 mL/min using a Superose 6–10/300-GL column (GE Biosciences, Silverwater, Australia) equilibrated in PBS.

### 2.5. 4,4′-Dianilino-1,1′-binaphthyl-5,5′-disulfonic Acid (Bis-ANS) Assay

PAI-2^G^ or PAI-2^NG^ were incubated (37 °C, 30 min) at 0.05 mg/mL with 20 µg/mL bis-ANS. The protein solutions were dispensed into 384-well plates (*n* = 3, 50 µL/well), and blank-corrected bis-ANS fluorescence (excitation = 355 nm, emission = 480 nm) was determined using a POLARstar OMEGA plate reader (BMG Labtech, Melbourne, Australia).

### 2.6. Dityrosine Fluorescence and Circular Dichroism (CD) Measurements

PAI-2^G^ or PAI-2^NG^ were diluted to 0.1 mg/mL after dialysis in 10 mM sodium phosphate buffer (pH 7.4). The intrinsic fluorescence of the samples (excitation = 325 nm, emission = 420 nm), indicative of dityrosine cross-linkages as a surrogate marker of oxidative stress [23], was measured using a JASCO FP-8300 spectrofluorometer (ATA Scientific Instruments, Sydney, Australia). For CD spectroscopy, samples were analysed across 190–250 nm at 24 °C using a JASCO J-810 spectropolarimeter (ATA Scientific Instruments).

### 2.7. SDS-PAGE

Proteins were diluted in 4× denaturing loading buffer (200 mM Tris, 40% (*v*/*v*) glycerol, 2% (*w*/*v*) SDS, 0.4% (*w*/*v*) bromophenol blue, pH 6.8) and electrophoresed using 1.5 mm NuPAGE Bis-Tris 4–12% gels (Thermo Fisher Scientific) in 2-(N-morpholino)ethanesulfonic acid (MES)-SDS running buffer (50 mM MES, 50 mM Tris, 0.1% (*w*/*v*) SDS, 1 mM EDTA, pH 7.3) at 200 V for 35 min in an Invitrogen Mini Gel Tank electrophoresis system (Thermo Fisher Scientific, Waltham, MA USA). Protein size was estimated using Precision Plus Dual Colour Protein Standards (Bio-Rad, Gladesville, Australia). All gels were stained overnight with Instant Blue (Expedeon, Cambridge, UK) and subsequently destained in MilliQ H_2_O overnight.

### 2.8. Native PAGE

Native Tris-glycine gels (8%) were hand-cast for native PAGE. Briefly, 10 mL of 8% resolving polyacrylamide gel solution was prepared (4.7 mL MilliQ H_2_O, 2.7 mL 30% (*w*/*v*) acrylamide/bisacrylamide (37.5:1), 2.5 mL Tris (1.5 M, pH 8.8), 0.1 mL 10% (*w*/*v*) ammonium persulfate, with 0.006 mL tetramethylethylenediamine (TEMED) added directly prior to gel casting) and pipetted into 1.5 mm spaced glass plates in a Mini-PROTEAN Tetra Cell casting system (Bio-Rad,). The resolving gel was then overlaid with 3 mL of 5% stacking polyacrylamide gel solution. Proteins were diluted in 4× native loading buffer (200 mM Tris, 40% (*v*/*v*) glycerol, 0.01% (*w*/*v*) bromophenol blue, pH 8.6), and subsequently electrophoresed at 100 V for 1.5 h in native Tris-glycine running buffer (25 mM Tris, 192 mM glycine, pH 8.3) using a Mini-Protean electrophoresis system (Bio-Rad). Gels were stained as before.

### 2.9. uPA Activity Assay

The uPA protease inhibitory activity of PAI-2^G^, PAI-2^NG^ or PAI-1 was conducted and analysed as previously described [24]. Briefly, 0.25 mM Z-Gly-Gly-Arg-AMC fluorogenic uPA substrate (Calbiochem, Bayswater, Australia) in HEPES buffer (20 mM HEPES, 100 mM NaCl, 0.5 mM EDTA, 0.01% (*v*/*v*) Tween-20, pH 7.6), that was supplemented with 0, 1 or 2 nM PAI-2^G^, PAI-2^NG^ or PAI-1 pre-treated with 0, 100 or 200 μM NaOCl, was added to a black 96-well plate (*n* = 3, 100 μL/well). Upon subsequent addition of 1 nM uPA (Molecular Innovations) to the wells (100 μL), the fluorescence (excitation = 355 nm, emission = 460 nm) was immediately measured at 37 °C in 30 s cycles for 5 h using a POLARstar OMEGA plate reader. Measurements were adjusted using a uPA fluorogenic substrate/HEPES buffer blank (*n* = 3, 100 μL/well).

### 2.10. Thioflavin-T (ThT) Assay

Aβ_1–42_ (3 μM in PBS containing 25 μM ThT) was added to a clear 384-well plate (*n* = 3, 80 µL/well) in the presence or absence of PAI-2^G^ or PAI-2^NG^ (pre-treated with 0–200 µM NaOCl). The plate was incubated at 28 °C with periodic shaking, and ThT fluorescence (excitation = 440 nm, emission = 480 nm) was measured using a POLARstar OMEGA plate reader. SERPINB14 (ovalbumin), a non-chaperone clade B serpin that shares high sequence homology with PAI-2 was used as a control in these experiments.

### 2.11. Western Blot

Samples from ThT assays (*n* = 3, 80 µL/well) were pooled, centrifuged at high speed (24,000× *g*, 20 min, room temperature), and 30% of the supernatant volume was subsequently separated by native PAGE. Protein samples were then transferred to a 0.45 µm nitrocellulose membrane (Pall Corporation, Somersby, Australia) via electrophoresis at 100 V for 90 min using a Mini Trans-Blot electrophoresis system (Bio-Rad) containing Western transfer buffer (192 mM glycine, 25 mM Tris, 20% (*v*/*v*) methanol, pH 8.3). Membranes were then blocked overnight at 4 °C in PBS containing 10% (*w*/*v*) skim milk powder. Membranes were subsequently probed using either a rabbit polyclonal anti-PAI-2 antibody (Abcam, Melbourne, Australia; ab137588) or monoclonal anti-Aβ antibody clone W0-2 (Millipore, Sydney, Australia; MABN10). Typically, membranes were incubated with the primary antibody for 1 h at room temperature with gentle shaking, and then washed for 10 min with PBS containing 0.01% (*v*/*v*) Triton X-100 and then twice for 10 min with PBS only. The binding of the primary antibodies was detected by subsequent incubation of the membrane with highly cross-adsorbed anti-mouse (GE Biosciences; NXA931) or anti-rabbit (Thermo Fisher Scientific; A16194) horseradish peroxidase (HRP)-conjugated secondary antibodies, for 30 min at room temperature with gentle shaking. Finally, the membranes were washed as before, and bands were visualised by enhanced chemiluminescence using Clarity substrate (Bio-Rad) and an Amersham 600RGB Imager (GE Biosciences). All antibodies were used at concentrations of 1:1000, diluted in PBS containing 10% (*w*/*v*) skim milk powder.

### 2.12. Aβ_1–42_ Cytotoxicity Assays

Before commencement of cytotoxicity assays, all protein solutions were prepared in PBS (pH 7.4) and 0.22 μm sterile-filtered. SH-SY5Y neuroblastoma cells (ATCC CRL-2266) were initially seeded at 40% confluency in a 96-well plate with DMEM/F-12 media supplemented with 10% (*v*/*v*) FCS (Bovogen Biologicals, Melbourne, Australia). The next day, the media were aspirated and carefully refreshed with treatments consisting of Aβ_1–42_ (10 μM) prepared in neurobasal medium/1× B-27/1× GlutaMAX ± PAI-2^G^, PAI-2^NG^ or SerpinB14 at a 1:10 molar ratio of PAI-2/SerpinB14:Aβ_1–42_ (*n* = 3 technical replicates/100 μL/well). Vehicle control cells were also prepared that received equivalent volumes of growth medium with PBS instead of protein solution (“medium alone”). All protein solutions were incubated for 5 h at 37 °C prior to being added to SH-SY5Y cells, to generate toxic soluble oligomers of Aβ_1–42_ Upon addition of the protein solutions, all wells were immediately supplemented with 250 nM Cytotox Green dead cell stain (Essen Bioscience, Melbourne, Australia; according to the manufacturer’s instructions) and further incubated under standard cell culture conditions for 48–96 h once maximal Aβ_1–42_ cytotoxicity had been reached. Cells were visualised using an IncuCyte ZOOM (2016A) live-cell imager (Essen Bioscience, Melbourne, Australia) on the phase and green fluorescence (excitation = 490 nm, emission = 524 nm; acquisition time = 400 ms) channels fitted with a 10× objective. Fluorescent object count (FOC; indicative of cell death) was calculated using an IncuCyte green object analysis mask under a fixed threshold with a green calibration unit (GCU) of 100 and edge split turned on (set to 0). All treatments were converted to cytotoxicity (%) relative to the Aβ_1–42_ alone treatment, taken as 100% cytotoxicity.

### 2.13. Immunohistochemistry and Immunofluorescence

Placenta was collected from patients with deliveries in the context of early-onset severe preeclampsia (sPE, *n* = 15 gestational age (GA) at delivery (mean ± standard deviation): 30 ± 2 weeks) or spontaneous idiopathic preterm birth at similar gestational ages (iPTB, *n* = 15, GA: 30 ± 2 weeks) as best possible control. All subjects signed informed consent approved by the Human Investigation Committee of Yale University (Protocol 2019-0290). A summary of demographic and clinical characteristics is shown in Appendix A.

A full-thickness biopsy was retrieved from the central portion of the placenta within minutes from the delivery, fixed in formalin and subsequently embedded in paraffin. Serial sections (5 µm) were deparaffinised, rehydrated and subjected to antigen retrieval with citrate buffer. Sections were treated sequentially with 1% hydrogen peroxide for 15 min, followed by 1 h incubation with 5% donkey serum (Jackson ImmunoResearch Laboratories, West Grove, PA, USA) and then incubated overnight at 4 °C with primary antibodies, rabbit polyclonal anti-PAI-2 or monoclonal anti-Aβ antibody clone W0-2. Detection and signal amplification was performed with biotinylated donkey anti-rabbit or anti-mouse IgG (1:600, Jackson ImmunoResearch Laboratories) and avidin-biotin (VECTASTAIN^®^ Elite ABC, Vector Laboratories, Burlingame, CA, USA) using Vector NovaRed as peroxidase substrate. Sections exposed to non-immune IgG served as negative control. Intensity of staining in syncytiotrophoblast was appreciated semiquantitatively on a scale from 0 to 5 by an observer blinded to clinical grouping. Scores from three distinct fields were averaged for each case. Concordance between the observers was 0.874 ± 0.045 Cohen’s kappa index, 95% CI: 0.786 to 0.962.

To evaluate the possibility that PAI-2 co-localises with β-amyloid, double immunofluorescence on select tissues was performed. After deparafinisation and antigen retrieval with citrate buffer, slides were blocked with 100 mM glycine followed by 10% goat serum for 1 h at room temperature. Slides were incubated overnight at 4 °C with a cocktail of anti-PAI-2 (1:500) and W0-2 antibody (1:250; recognises amino acid residues 4–10 of human Aβ) followed by 1 h incubation at room temperature in secondary antibody cocktail (2 µg/mL goat anti-mouse IgG-Alexa Fluor 488, 2 µg/mL goat-anti-rabbit IgG-Alexa Fluor 594 and 1 µg/mL DAPI). Slides were mounted with ProLong Gold Antifade medium and images captured using a Nikon Eclipse Ti inverted microscope (Nikon Instruments, Melville, NY, USA).

## 3. Results

Under native conditions, approximately 45% of PAI-2^G^ and 50% of PAI-2^NG^ eluted at positions corresponding to monomers by SEC (Figure 1A,B). SDS-PAGE analysis confirmed the size of monomeric PAI-2^G^ as ~60 kDa and PAI-2^NG^ as ~47 kDa (Appendix A). Hypochlorite treatment induced PAI-2^G^ and PAI-2^NG^ to form higher-order assemblies in a dose-dependent manner; however, PAI-2^NG^ typically formed larger assemblies (>360 kDa) compared to PAI-2^G^ (Figure 1A,B). Similar results were obtained by native PAGE analysis (Appendix A). There was a dose-dependent increase in dityrosine fluorescence following hypochlorite treatment of both PAI-2 forms (Figure 1C), which was slightly greater for PAI-2^NG^ vs. PAI-2^G^ at 200 µM NaOCl. CD spectroscopy analysis revealed that the secondary structures of PAI-2^G^ and PAI-2^NG^ were comparable under native conditions (Figure 1D). Both forms of PAI-2 were largely resistant to hypochlorite-induced misfolding but decreased in minima at 208 nm and 222 nm, indicating a small reduction in α-helical content following treatment with hypochlorite (Figure 1E,F).

It has been reported that the protease inhibitory activity of PAI-2^NG^ is resistant to inactivation by hypochlorite [10]. However, a direct comparison of this property of PAI-2^NG^, to that of PAI-2^G^ or PAI-1, has not been performed previously. As expected, under native conditions, all PAIs virtually abolished the activity of uPA at a molar ratio of 1:1 or 2:1 PAI:uPA (Appendix A). Hypochlorite pre-treatment, in a dose-dependent manner, reduced the ability of all PAIs to inhibit uPA at a 1:1 molar ratio (Figure 2A,B and Appendix A). However, in terms of their sensitivity to hypochlorite-induced inactivation, PAI-1 > PAI-2^G^ > PAI-2^NG^. This trend was more obvious when the PAIs were pre-treated using 100 μM NaOCl compared to the lower concentration of 50 μM NaOCl (Figure 2A,B). The same trend was evident when the assay was repeated at a 2:1 molar ratio of PAI:uPA (Appendix A).

As assessed by ThT assay, PAI-2^G^ and PAI-2^NG^ inhibited the aggregation of Alzheimer’s disease- and preeclampsia-associated Aβ_1–42_ by extending the initial lag phase (during which there is no increase in ThT fluorescence) and significantly decreasing the rate at which ThT fluorescence increased in a dose-dependent fashion (Figure 3A,B; Appendix A). Comparatively, this activity was more efficient for PAI-2^G^ compared to PAI-2^NG^ (Appendix A). Following prolonged co-incubation (i.e., 12 h) of PAI-2^NG^ with a high-molar excess of Aβ_1–42_, the maximum ThT fluorescence could reach values higher than that of control samples containing Aβ_1–42_ alone (Figure 3D and Appendix A). However, this difference was not always statistically significant due to the variability associated with high fluorescence measurements (Figure 3B). PAI-2^G^ was also substantially more effective than PAI-2^NG^ at reducing the initial rate of Aβ_1–42_ aggregation, when compared at an equivalent mass ratio (Appendix A). Interestingly, PAI-2^G^ (but not PAI-2^NG^) inhibited the amorphous aggregation of creatine phosphokinase in a dose-dependent manner (Appendix A).

Under the conditions used, hypochlorite pre-treatment did not have a significant effect on the chaperone activity of PAI-2^G^ (Figure 3C, Appendix A), but dramatically increased the ability of PAI-2^NG^ to inhibit the rate at which amyloid-associated ThT fluorescence increased and the maximum ThT fluorescence reached during the assay (Figure 3D; Appendix A). Consistent with these findings, at the conclusion of the assay, Aβ_1–42_ co-migrated with hypochlorite-induced high-molecular-mass PAI-2^NG^ species as assessed by native Western blot analysis (Figure 3E). Interestingly, high-molecular-mass PAI-2^NG^ and PAI-2^G^ species were induced following hypochlorite treatment (Figure 3E). Considering that native PAI-2^G^ efficiently inhibited Aβ_1–42_, the data support the idea that either the stoichiometry of stable complexes formed between PAI-2 and Aβ_1–42_ is strongly biased towards a greater amount of Aβ_1–42_ or that transient interactions also contribute to the stabilising effect of PAI-2 on Aβ_1–42_. SerpinB14, a clade B serpin like PAI-2, had a negligible effect on Aβ_1–42_ aggregation following 0–100 µM NaOCl treatment (Appendix A), indicating a specific effect of the PAI-2 forms. Analysis of the soluble protein fraction present following prolonged co-incubation of PAI-2 with Aβ_1–42_ at a 1:5 molar ratio showed that all forms of PAI-2 enhance the solubility of Aβ_1–42_ and the formation of stable PAI-2-Aβ_1–42_ complexes, as assessed using native Western blotting and biotin/streptavidin pull-down assays, respectively (Appendix A). Pre-incubation of PAI-2^G^ or PAI-2^NG^ with Aβ_1–42_ at a 1:10 molar ratio protected SH-SY5Y cells against toxic Aβ_1–42_ species by around 65% and 43%, respectively (Figure 3F and Appendix A). The cytoprotective activity of both PAI-2 forms was resistant to hypochlorite pre-treatment, producing similar protection to those of non-oxidised PAI-2 forms under the conditions used (Figure 3F). Pre-incubation of Aβ_1–42_ with SerpinB14 had no effect on the toxicity of Aβ_1–42_ in vitro (Figure 3F), which is consistent with the idea that the cytoprotective effect is linked to the chaperone activity of PAI-2.

To provide evidence for their co-localisation in vivo, we stained PAI-2 and APP (which is cleaved to form Aβ) in the placenta of women with sPE, which are known to harbour plaque-like aggregates. PAI-2 primarily localised to syncytiotrophoblast within the villous placenta (Figure 4A,B). Although a wide interpersonal variation in staining intensity was noted in both iPTB controls (Figure 4A) and sPE placentae (Figure 4B), one pattern stood out for some sPE cases whereby pleomorphic PAI-2-stained vesicles appeared to abundantly extrude out from the syncytiotrophoblast into the maternal vascular spaces. In contrast, PAI-2 staining was absent in control slides incubated with non-immune IgG instead of PAI-2 antibody (Figure 4C). Among sPE cases, those with manifestations of haemolysis, elevated liver enzymes and low platelets (HELLP) syndrome tended to have low placental PAI-2 staining scores (Figure 4D); however, staining of PAI-2 was not significantly different between iPTB and sPE placentas in this study. By immunofluorescence analysis, the majority of amyloid precursor protein (APP, which is cleaved to form Aβ) and PAI-2 are discretely localised; however, some co-localisation of APP and PAI-2 within cells and in extracellular deposits was evident (Figure 4E,F).

## 4. Discussion

Our findings provide support for a novel function of PAI-2 as a hypochlorite-resistant protease inhibitor and holdase-type chaperone. It is plausible that this feature of PAI-2 is paramount to its upregulation in pregnancy, which is associated with chronic oxidative stress [18,25,26], and in pathological inflammatory states in which the generation of reactive oxygen species, including hypochlorite, can reach millimolar levels [27,28,29]. Although secreted PAI-2^G^ is considered the primary extracellular form, PAI-2^NG^ can be exported from leukocytes and endothelial cells in response to inflammatory stimuli [30,31]. Therefore, it is interesting that PAI-2^NG^ is relatively more resistant to hypochlorite-induced inactivation of its protease inhibitory activity compared to PAI-2^G^ and that the chaperone activity of PAI-2^NG^, but not PAI-2^G^, is enhanced by hypochlorite-induced modification. Supporting the conclusion that PAI-2 is critically important in controlling fibrinolysis and extracellular proteostasis during inflammation, SerpinB2 knockout mice experience defective kidney repair following inflammatory damage [32], increased sensitivity to topical inflammatory agents [33] and accelerated tumour growth (i.e., a process driven by the inflammatory microenvironment) [34]. Additionally, reduced expression of PAI-2 has been implicated in preeclampsia in humans [19,20,21], a major cause of maternal and neonatal morbidity and mortality, that involves high levels of inflammation, oxidative stress, aberrant fibrinolysis and accumulation of misfolded proteins [22,35,36]. Although there was large interpersonal PAI-2 staining variation, our immunofluorescence data provides the first evidence that a fraction of endogenous PAI-2 is found co-located with APP (or aggregation-prone proteoforms) in the placenta of preeclampsia patients, which is consistent with the notion that PAI-2 may preferentially bind to misfolded APP proteoforms in vivo.

Previous research has shown that intracellular PAI-2^NG^ attenuates the aggregation of numerous misfolded proteins, promotes the formation of cytoprotective inclusion bodies and binds to the proteasome in vitro [9,37]. Here, we show that this is a concentration-dependent effect. In the presence of a high-molar excess of Aβ_1–42_, PAI-2^NG^ can also promote Aβ_1–42_ aggregation after prolonged incubation in vitro. Similar to the mechanism proposed for other extracellular chaperones [38], this could be due to preferential stabilisation of Aβ_1–42_ assemblies formed early on the amyloid-forming pathway when PAI-2 is present at limited concentration. Prior studies of the importance of glycosylation on holdase-type chaperone activity have reported conflicting data [14,39]. This is potentially due to the methods used to generate deglycosylated protein, which can result in incomplete removal of carbohydrate groups and denaturation of the protein [14,39]. By comparing the activity of recombinant PAI-2 generated by bacterial or mammalian expression systems, our data support the conclusion that carbohydrate groups enhance the chaperone activity of PAI-2 by promoting the formation of chaperone-active oligomeric assemblies.

Similar to intracellular small heat shock proteins [40] and the well-characterised extracellular chaperone clusterin [41] and alpha2-macroglobulin [42], PAI-2 appears to stabilise misfolded proteins in very high-molecular-mass complexes. That hypochlorite-induced modification preferentially induces PAI-2^NG^ to form very high-molecular-mass oligomers compared to PAI-2^G^ (possibly because glycosylation prevents the non-covalent association of hydrophobic regions of PAI-2); potentially underlies the reason for hypochlorite-induced enhancement of PAI-2^NG^ chaperone activity. However, elevated dityrosine bond formation might also be a contributing factor. While hypochlorite-induced modification did not enhance the cytoprotective effect of PAI-2 against Aβ_1–42_ neurotoxicity it is important to note that ThT-positive Aβ_1–42_ species are heterogeneous and do not correlate directly with measures of cytotoxicity [43]. Additionally, it is plausible that hypochlorite-induced modification disrupts molecular interactions that are important to the cytoprotective effect of PAI-2, such as its binding to cell surface receptors. Previous studies have shown that PAI-2 is reactive in microglia that surround amyloid plaques in Alzheimer’s disease [44], supporting the notion that both its serpin and chaperone activities may be important in the local inflammatory microenvironment. It is tempting to speculate that PAI-2 performs these functions in the placenta, which is also known to (i) accumulate hypochlorite-induced protein damage during ageing [18] and (ii) express many receptors implicated in Aβ-induced toxicity [45,46,47,48]. However, it has yet to be demonstrated that Aβ is directly toxic to placental trophoblast cells. An alternative hypothesis is that Aβ contributes to placental dysfunction in preeclampsia by impairing endothelial cells [49] or stimulating immune cells [50].

The uPA inhibitory activity of all PAIs was sensitive to hypochlorite; however, the magnitude of the effect differed markedly between the proteins with PAI-2^G^ and particularly PAI2^NG^, being significantly more resistant to hypochlorite inactivation compared to PAI-1. PAI-1 sensitivity to hypochlorite is likely due to the presence of a highly oxidisable methionine residue in the P1′ inhibitory site [10,51]; however, it has also been reported that perturbed secondary structure, independent-from methionine oxidation, is the critical factor that inactivates PAI-1 [11]. Although PAI-2^G^ is heavily glycosylated, our data suggest that this only modestly shields PAI-2 from hypochlorite-induced dityrosine formation and concurrently promotes secondary structure perturbations. This, or the formation of loop-sheet oligomers, which are non-inhibitory [52], may explain the loss of serpin-related activity induced by hypochlorite, especially for PAI-2^G^. Considering the relative sensitivity of PAIs to hypochlorite-induced inactivation, it is possible that PAI-2 could become the dominant inhibitor of uPA and tPA during inflammatory conditions featuring enhanced oxidative stress, such as atherosclerosis [53,54,55,56], periodontitis [57,58,59] and particularly pregnancy [35,60,61], when there is marked upregulation of PAI-2 at the placenta and in blood plasma.

A large number of maternal adaptations are necessary for successful pregnancy and to ensure the health of the mother. Our data support the conclusion that induction of PAI-2 is a response that supplements the existing network of molecular machinery to control fibrinolysis and extracellular protein misfolding locally at the placenta. Additionally, it is feasible that upregulation of PAI-2 is called upon to perform these activities in response to inflammation in other tissues. Further studies are needed to define the relative contribution of PAI-2 controlling fibrinolysis and protein misfolding during relevant physiological scenarios in vivo.

In conclusion, this study forms part of a growing body of work characterising how the proteome is exquisitely responsive to hypochlorite-induced oxidative stress [42,62,63]. A more detailed understanding of molecular adaptations to oxidative stress could help to reveal novel therapeutic strategies for controlling proteostasis in a broad range of inflammatory diseases.

## Figures and Tables

**Figure 1 cells-11-01152-f001:**
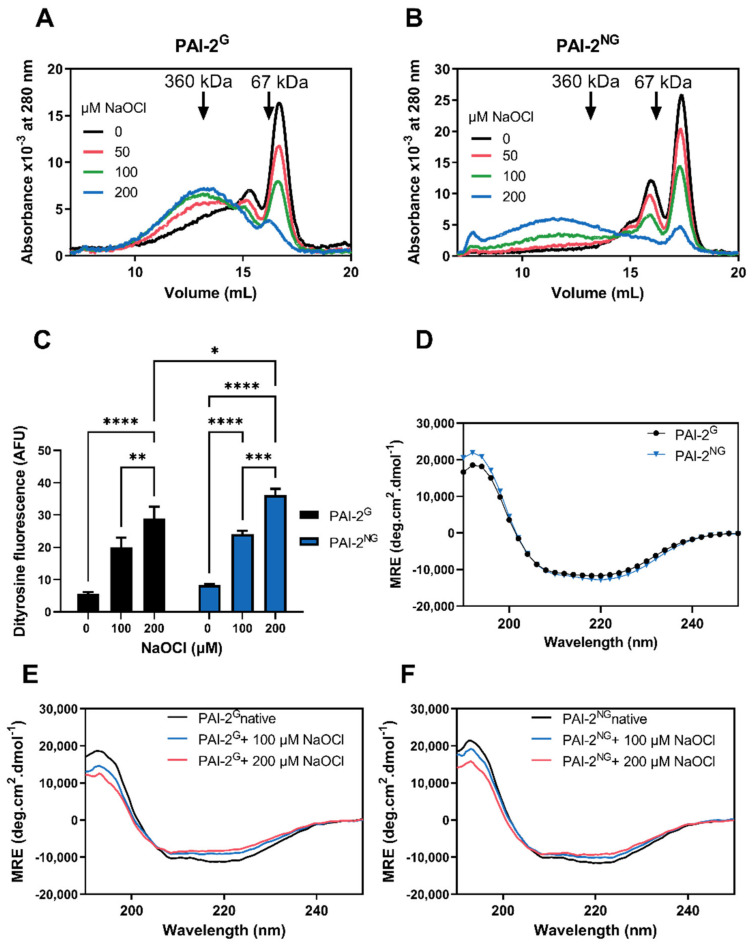
Biochemical analyses showing the effect of hypochlorite on PAI-2^G^ and PAI-2^NG^. Size exclusion chromatography showing hypochlorite-induced oligomerisation of (**A**) PAI-2^G^ and (**B**) PAI-2^NG^. Arrows depict elution of molecular weight standards pregnancy zone protein (360 kDa) and bovine serum albumin (67 kDa). (**C**) PAI-2^G^ or PAI-2^NG^ pre-treated with 0–200 μM hypochlorite were analysed for intrinsic fluorescence (AFU; excitation = 325 nm, emission = 420 nm). Data are means ± SD (*n* = 3) and are representative of three independent experiments. One-way ANOVA with post-hoc Tukey’s HSD was used to identify non-significant (ns) and significant differences within the PAI-2 forms compared to untreated controls (**** = *p* < 0.0001; *** = *p* < 0.001; ** = *p* < 0.01; * = *p* < 0.05). There were no significant differences between untreated and treated PAI-2^G^ or PAI-2^NG^ except for the 200 μM treatment; * *p* = 0.037. (**D**) CD spectroscopy displayed as mean residue ellipticity (MRE) of PAI-2^G^ and PAI-2^NG^ measured between 190 and 250 nm at 24 °C under native conditions or following 0–200 μM hypochlorite pre-treatment of (**E**) PAI-2^G^ or (**F**) PAI-2^NG^.

**Figure 2 cells-11-01152-f002:**
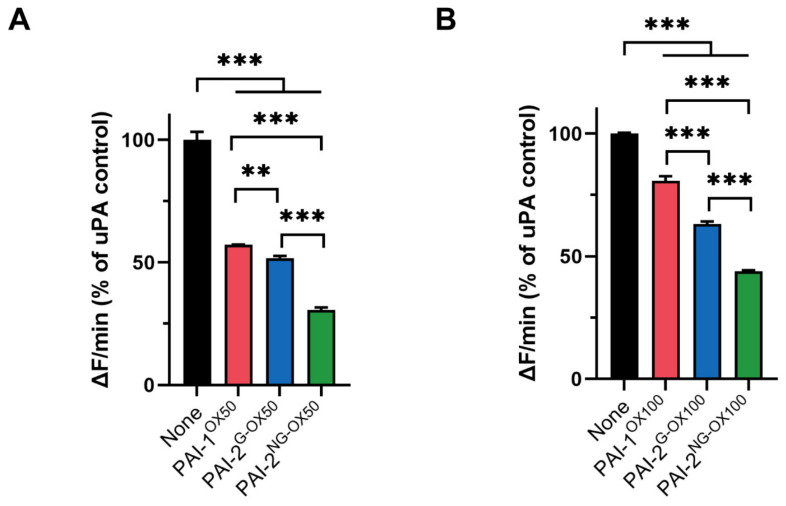
The uPA inhibitory activities of PAI-2^G^ and PAI-2^NG^ are more resistant to hypochlorite than PAI-1. PAIs pre-treated with either (**A**) 50 µM NaOCl (OX50) or (**B**) 100 µM NaOCl (OX100) were added to the uPA fluorescence substrate containing uPA at a 1:1 molar ratio. None = uPA in buffer alone, no PAIs added. Change in fluorescence/min (ΔF/min) for each treatment over 1 h was converted to a percentage of the maximal (uninhibited) uPA activity. Values shown are the means ± SD (*n* = 3) from a representative experiment repeated 3 times. One-way ANOVA with post-hoc Tukey’s HSD was used to identify non-significant (ns) and significant differences (*** = *p* < 0.001; ** = *p* < 0.01). Kinetic curve data are presented in Appendix A.

**Figure 3 cells-11-01152-f003:**
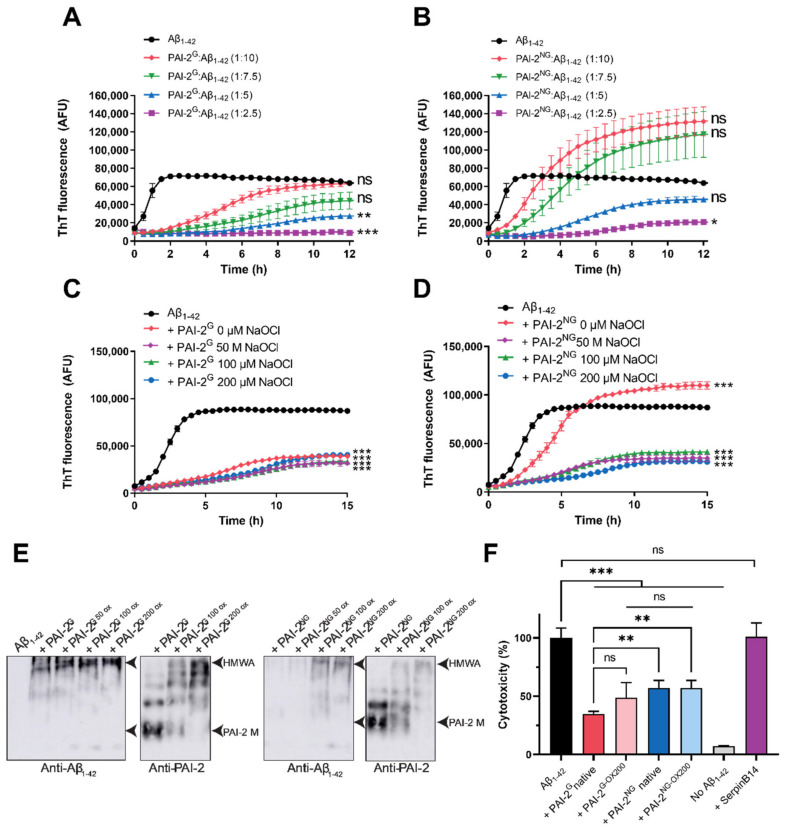
The effect of native or hypochlorite-treated PAI-2^G^ and PAI-2^NG^ on the aggregation and cytotoxicity of Aβ_1–42_. (**A**,**B**) Aβ_1–42_ (3 µM) was incubated with 25 µM ThT in PBS at 28 °C ± (**A**) PAI-2^G^ or (**B**) PAI-2^NG^ at the molar ratios indicated (1:2.5–10 PAI-2:Aβ_1–42_). (**C**,**D**) Aβ_1–42_ was incubated as before with (**C**) PAI-2^G^ or (**D**) PAI-2^NG^, which had been pre-treated with 0–200 µM hypochlorite. The molar ratio of PAI-2 to Aβ_1–42_ is 1:10 and 1:7.5 in panels C and D, respectively. All data are means ± SEM (*n* = 3) from a representative experiment repeated three times. One-way ANOVA with post-hoc Tukey’s HSD was conducted on all ThT assay endpoints to identify non-significant (ns) and significant differences (*** = *p* < 0.001; ** = *p* < 0.01; * = *p* < 0.05) with respect to the Aβ_1–42_ alone control. Additional statistical analysis is presented in Appendix A. (**E**) Samples taken at the conclusion of ThT assays in (**C**,**D**) were centrifuged and the supernatants separated by native PAGE using 8% Tris-glycine gels. After blotting, the samples were probed with either anti-Aβ_1–42_ or anti-PAI-2 antibodies. The position of soluble high molecular weight aggregates of Aβ_1–42_ (HMWA) and PAI-2 monomers (PAI-2 M) are marked on each blot. (**F**) SH-SY5Y cells were treated with neurobasal medium containing Aβ_1–42_ (10 µM) ± PAI-2^G^ or PAI-2^NG^ pre-treated with 0 or 200 µM NaOCl (OX200), or native SerpinB14, at a 1:10 molar ratio of PAI-2/SerpinB14:Aβ_1–42._ Cells were monitored over 48–96 h (maximal toxicity). Neurobasal medium (no Aβ_1–42_) was included as a control. Data are means ± SD (*n* = 6) from two independent biological experiments. Statistical analysis was conducted as before.

**Figure 4 cells-11-01152-f004:**
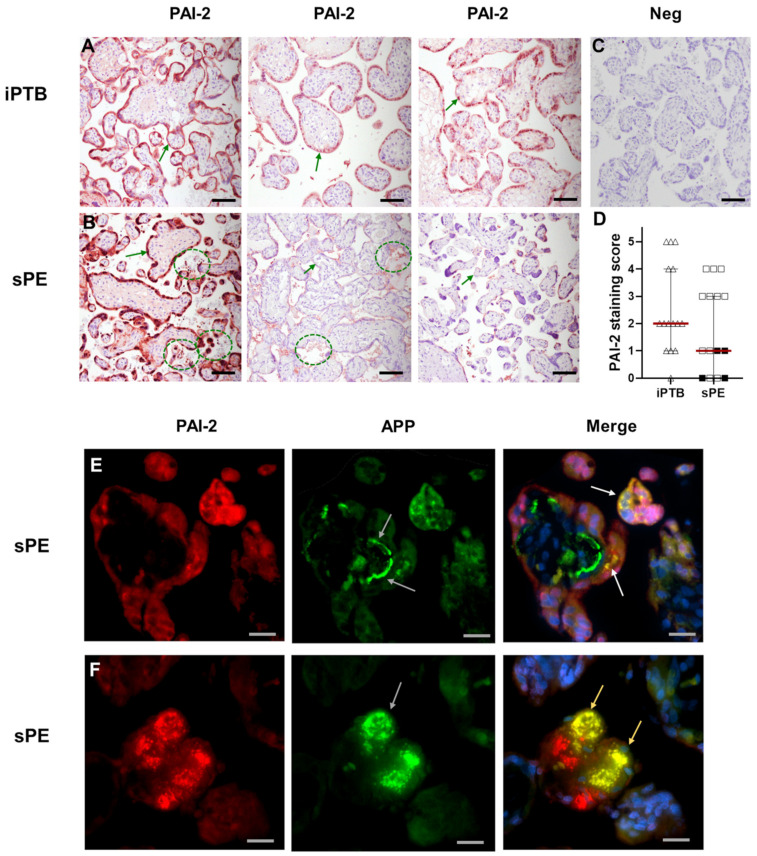
Immunolocalisation of PAI-2 and APP in placenta. Representative immunostaining for PAI-2 in placenta of women who delivered in the context of idiopathic preterm birth (iPTB, (**A**)) or severe early-onset preeclampsia (sPE, (**B**)). Green arrows mark the syncytiotrophoblast. Each micrograph is from a different case to illustrate the interpersonal variation in staining pattern in both groups. The marquee-circled areas ((**B**), **left** and **middle** images) show PAI-2 positive debris abundant in maternal vascular spaces observed frequently in sPE cases. Also in panel (**B**) (**right** image) is the placenta from women with sPE and HELLP syndrome, which shows only weak staining (score 1). The negative image ((**C**); Neg) is from a slide incubated with non-immune IgG instead of primary antibody. (**A**–**C**), original magnification 200× (scale bar: 50 μm). (**D**) Scatterplot of PAI-2 staining score for iPTB (*n* = 15) and sPE (*n* = 15). sPE women with HELLP manifestations (*n* = 4) are shown as filled black squares. (**E**,**F**), Double immunofluorescence for PAI-2 (red) and APP (green) on a representative sPE placenta, which shows that some but not all PAI-2 staining co-localises with APP. The white arrows in (**E**) (merge image) point to intracellular co-localisation of PAI-2 and APP in syncytiotrophoblast. Note the irregular shape of extracellular APP staining shown by grey arrows ((**E**,**F**), APP images), consistent with aggregated Aβ, which is free of PAI-2 staining in (**E**) (merge image), but partially co-stains with PAI-2 as marked by yellow arrows in (**F**) (merge image). The blue counterstain marks the nuclei. (**E**,**F**) original magnification 400× (scale bar: 25 μm).

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
