# Peer review of "A Novel Role for Plasminogen Activator Inhibitor Type-2 as a Hypochlorite-Resistant Serine Protease Inhibitor and Holdase Chaperone"

_cells, 2022, doi:10.3390/cells11071152_

Round 1
Reviewer 1 Report
The authors have responded satisfactorily to my earlier comments. Thank you
Author Response
Thank you
Marie
Reviewer 2 Report
The authors have satisfactorily addressed all of the reviewer's comments and suggestions. The revised manuscript is scientifically sound and acceptable. Additional data was also helpful in evaluating the strength of the manuscript.
Reviewer 3 Report
The authors have provided sufficient answers to the comments I have posed.
Minor comment:
In Figure 1C, authors used one-way ANOVA with post-hoc Tukey's HSD and show p-values of post-hoc analysis between two groups, but the comparison target two groups were not clearly marked in ***. Please clearly indicate comparison groups of *** in Figure 1C.
Author Response
We have now redrawn Figure 1C for easier interpretation of differences between treatments on each PAI-2 form and clearly highlighted where there was a significant difference between the PAI-2 forms due to treatment (in this case 200uM NaOCl). We also reworded the relevant part of Figure 1 legend to reflect this. This is shown as a track change in the revised version.
Reviewer 4 Report
I found this article to be a very interesting and one that is certainly worthy of publication in Cells.
Author Response
We thank the Reviewer for their positive review of our manuscript.